# Trauma Can Induce Telangiectases in Hereditary Hemorrhagic Telangiectasia

**DOI:** 10.3390/jcm9051507

**Published:** 2020-05-17

**Authors:** Urban Geisthoff, Ha-Long Nguyen, Rolf Lefering, Steffen Maune, Kruthika Thangavelu, Freya Droege

**Affiliations:** 1Department of Otorhinolaryngology, Head and Neck Surgery, University Hospital Marburg, Philipps-Universität Marburg, Baldingerstrasse, 35043 Marburg, Germany; geisthof@med.uni-marburg.de (U.G.); kruthika.thangavelu@gmail.com (K.T.); 2Morbus Osler-Selbsthilfe e.V. (German HHT Self-Help Group), 89264 Weissenhorn, Germany; 3Laboratory of Human Molecular Genetics, de Duve Institute, Université catholique de Louvain, 1200 Brussels, Belgium; ha-long.nguyen@uclouvain.be; 4Institute for Research in Surgical Medicine (IFOM), University of Witten/Herdecke, Campus Merheim, 51109 Cologne, Germany; rolf.lefering@uni-wh.de; 5Department of Otorhinolaryngology, Hospitals of the City of Cologne, University of Witten/Herdecke, 51067 Cologne, Germany; maunes@kliniken-koeln.de; 6Department of Otorhinolaryngology, Head and Neck Surgery, Essen University Hospital, University Duisburg-Essen, Hufelandstrasse 55, 45122 Essen, Germany

**Keywords:** hereditary hemorrhagic telangiectasia (HHT), telangiectases, mechanical damage, sun-induced trauma, vascular malformations, Endoglin, activin-receptor-like kinase 1

## Abstract

Hereditary hemorrhagic telangiectasia (HHT) is an autosomal dominant disease of the fibrovascular tissue resulting in visceral vascular malformations and (muco-) cutaneous telangiectases with recurrent bleedings. The mechanism behind the disease is not fully understood; however, observations from HHT mouse models suggest that mechanical trauma may induce the formation of abnormal vessels. To assess the influence of environmental trauma (mechanical or light induced) on the number of telangiectases in patients with HHT, the number of telangiectases on the hands, face, and lips were counted on 103 HHT patients possessing at least three out of four Curaçao criteria. They were then surveyed for information concerning their dominant hand, exposure to sunlight, and types of regular manual work. Patients developed more telangiectases on their dominant hand and lower lip (Wilcoxon rank sum test: *p* < 0.001). Mechanical stress induced by manual work led to an increased number of telangiectases on patients’ hands (Mann–Whitney U test: *p* < 0.001). There was also a positive correlation between sun exposure and the number of telangiectases on the lips (Mann–Whitney U test: 0.027). This study shows that mechanical and UV-induced trauma strongly influence the formation of telangiectases in HHT patients. This result has potential implications in preventive measures and on therapeutic approaches for HHT.

## 1. Introduction

Hereditary hemorrhagic telangiectasia (HHT) is a rare autosomal dominantly inherited disorder that affects the vasculature. The predominant vascular defects range from small telangiectases within nearly all cutaneous and mucocutaneous membranes to larger AV malformations (AVM) within the lungs, liver and brain. Diagnosis of HHT is established either by genetic testing or the fulfillment of at least three of the four Curaçao criteria (recurrent epistaxis, multiple telangiectases at characteristic sites, AVM in visceral lesions, and a family history) [1,2]. 

About 50–80% of HHT patients form (muco-)cutaneous telangiectases, predominantly on the mucosa of the nose and mouth, tongue, lips, face and fingers [1,2,3]. Since the dilated vessels are compromised, they are prone to rupture. Recurrent bleedings, especially epistaxis, can lead to severe anemia that can impair patients’ daily routine and quality of life [4,5,6]. An age-dependent penetrance of the disease [7] and highly variable clinical phenotypes are described. The age of onset, the severity and the location of the vascular lesions differ among each patient [8,9].

The mechanisms leading to new AVMs are not completely understood. In animal models for HHT mechanical stress like wounding or fluid shear stress induced new arteriovenous malformations [10,11]. However, to our knowledge, this has never been confirmed in adults with HHT. The aim of this study was to evaluate the influence of mechanical or sun-induced trauma on the number of telangiectases in patients with HHT.

## 2. Experimental Section

In 103 consecutive patients who fulfilled at least three out of four Curaçao criteria [1], telangiectases on both hands as well as the upper and lower lip were quantitated. Afterwards, they were surveyed about their dominant hand, level/type of regular manual labor involving strain on the hands (low, medium, high) and exposure to sunlight (low, medium, high). Examples were given to facilitate a graded assessment: medium strain by manual work would be a person from time to time but not daily having a wound on the hand due to mechanical stress from manual work. Medium sun exposure would be a person who frequently would get tanned but rarely had sunburns. Patients who have had laser therapy of their hands (one patient) or lips (17 patients) in the past were excluded from the analysis. 

Description of the study population included number of patients (n), mean ± standard deviation (SD), t test, and 95% confidence intervals (95% CI). A Pearson correlation coefficient was performed to analyze patients’ age and number of telangiectases. The Mann–Whitney U test and Kruskal–Wallis test were used for comparisons between patients, while paired tests (Wilcoxon rank sum; sign test) were used for intra-individual comparisons. A 5% significance level was determined. Statistical analyses were performed with IBM Corp. IBM SPSS Statistics for Windows, version 23.0. Armonk, NY, United States. 

### Study Approval

The authors assert that all procedures contributing to this work comply with the ethical standards of the Helsinki Declaration. The submission of the manuscript has been approved by the institutional review board (“Studienkommission”) of the hospitals of the city of Cologne (180920171130). Data was provided voluntarily by HHT patients. 

## 3. Results

The study cohort consisted of 61 females (59%) and 42 males (41%), ranging from age 9 to 83 years. No observable difference in the number of telangiectases was found between female and male patients (telangiectases on hands: men (m ± SD): 85 ± 126, women (m ± SD): 78 ± 94; Mann–Whitney U test: *p =* 0.536; telangiectases on lips: men (m ± SD): 15 ± 14, women (m ± SD): 19 ± 20; Mann–Whitney U test: *p =* 0.423). 

There was a positive correlation between advanced age and the number of telangiectases on hands (*r =* 0.429, *p <* 0.001). Patients developed more telangiectases on their dominant hand (dominant hand, mean: 47 telangiectases, 95% CI: 35–60; non-dominant hand, mean: 37 telangiectases, 95% CI: 27–47; Wilcoxon rank sum test: *p <* 0.001; see also Table 1 and Figure 1). Those patients were significantly older than those with telangiectases distributed equally on both hands and those with more telangiectases on the non-dominant hand (average age: more telangiectases on dominant hand (m ± SD): 55 ± 15 years, equally distributed: (m ± SD): 39 ± 16 years, more telangiectases on non-dominant hand: (m ± SD): 50 ± 17 years; Kruskal–Wallis test: *p =* 0.010). 

A typical example is shown in Figure 2. Additionally, patients reporting medium or high level of manual work in the past (*n =* 21, mean ± SD: 24 ± 34) showed significantly more telangiectases on their hands than patients with low levels of manual work (*n =* 77, mean ± SD: 7 ± 14; numbers add up to 98 as the four ambidextrous patients were excluded from the analysis) (Mann–Whitney U test: *p <* 0.01).

In evaluating the telangiectases on patients’ lips, 86 of the 103 patients were examined; 17 were excluded due to having had laser therapy on the lips in the past. Opposed to the findings in the hands, there was no statistically significant correlation between advanced age and the number of lip telangiectases, but only a tendency (*r =* 0.191, *p =* 0.078). More telangiectases were found on the lower lip in 75 patients and on the upper lip in two; three had an equal number on both lips, and six patients did not have any telangiectases (sign test: *p <* 0.001, lower lip: mean: 14 telangiectases 95% CI: 10–17; upper lip: mean: 4 telangiectases 95% CI: 3-5; Wilcoxon rank sum test < 0.001; numbers add up to 86 as 17 patients from 103 had laser therapy (Figure 3)). Of those 86 only 76 were able to describe their sun exposure sufficiently. It was also found that those reporting medium or high sun exposure (*n =* 9, mean ± SD: 14 ± 10) had more telangiectases on their lips than patients with low exposure to the sun (*n =* 67, mean ± SD: 9 ± 11) (Mann–Whitney U test: 0.027).

To emphasize the influence that the sun’s UV insult can have on formation of malformations in HHT, we note an anecdotal finding of a 65-year-old male patient who exhibited asymmetric distribution of telangiectases on his forehead. He informed us of his daily after-work ritual whereby he would sit on a bench on his balcony and read the newspaper. The sun would always shine on his left forehead, the side with more telangiectases (Figure 4). 

## 4. Discussion

The underlying pathogenic HHT mechanism remains unclear. The wide variability in age of onset and severity of symptoms amongst patients, even within the same family, suggests several factors contribute to the complexity of this disease. There does not appear to be gender propensity amongst HHT patients. Underscoring this point, no correlation between a patient’s sex and the number of telangiectases counted was found in this study. It has been observed that disorder progression worsens along with a patient’s age [12,13,14]. In accordance, we observed more telangiectases within older patients’ hands, and to some degree, lips. This could be due to the presumption that advanced aged patients have experienced more daily (muco-)cutaneous insults over time than younger patients. 

Patients are heterozygous for germline, loss-of-function mutations in certain members of the transforming growth factor-ß (TGF-ß) signaling pathway; most notably, endoglin (HHT1) or activin-like kinase receptor-1 (HHT2) [7,15,16]. It has been postulated that a homozygous state is not viable [17], which is supported by animal models [18,19]. It was initially presumed that haploinsufficiency of relevant HHT genes is responsible for development of vascular lesions. However, the unpredictable, focal nature of defects does not completely support this theory. Several other types of vascular malformations, such as inherited subsets of venous malformations and cerebral cavernous malformations, with a similar pattern of clinical manifestations were found to follow Knudson’s two-hit mechanism in which complete bi-allelic, localized loss of function of a gene of interest is required for formation of vascular lesions [20,21,22]. Recently, it was shown that HHT may also follow this trend rather than haploinsufficiency, as low-frequency somatic (mosaic) mutations were found in about half of the telangiectases isolated from HHT patients [23]. However, conditional mouse models in which both copies of ENG were deleted postnatally within endothelial cells also required a pro-angiogenic stimulus in order to develop vascular malformations. Conditional knockout models of ALK1 and SMAD4 formed vascular defects more consistently but inducing angiogenesis made lesions more robust. Hence, the mouse models suggest that loss of any of the HHT genes alone is not sufficient to form vascular lesions and that an external factor, such as in the form of wounding and shear/biomechanical stress, that triggers angiogenesis is needed [10,11,24,25,26,27,28,29,30,31]. A potential explanation is that since ALK-1 and ENG are increased in response to environmental and other physiological insults, such as fluid shear stress, vascular injury, inflammation, infection, ischemia, and angiogenic stimuli, the levels of wild type ALK-1 or ENG left is not sufficient for a vascular bed’s need to maintain homeostasis [11,32,33]. 

To our knowledge, this study is the first to confirm the influence that environmental factors play on lesion development in human patients. We assume that each patient in our cohort has a predisposing mutation in one of the HHT-associated genes; however, we cannot appropriately speculate the consequences of the loss of function as we did not have their genotypes. Our data in which more telangiectases were found on a patient’s dominant hand, and within patients whom performed more manual labor, indicates that triggers acting as an angiogenic stimulus play a vital role in the development of abnormal vessels in HHT. Patients in our cohort had more telangiectases on the lower, rather than the upper lip. One contributing factor could be that the lower lip is subjected to more physical insults than the upper lip; for example, it is in contact with the upper incisor teeth, or gravity naturally increases the chances of food touching the lower lip [34,35]. However, other environmental stimuli are likely be involved as well. The most probable cause is related to the fact that squamous cell carcinoma in the lower lip is more frequent than the upper lip. In oral oncology it is an accepted concept that the lower lip is more exposed to the sun. Additionally, surface area of the lower lip is larger in most individuals [36,37,38,39]. Similarly, HHT patients who claimed to have had excessive sun exposure in the past exhibited a higher number of telangiectases on the lips. In addition, one patient in the study displayed asymmetric distribution of telangiectases on his forehead, which corresponded to the side of his head that was predominantly exposed to the sun as he sat on his balcony. Thus, our data indicates that sunlight is able to stimulate development of new telangiectases; therefore, HHT patients should be advised to take extra precautions regarding sun protection when outdoors.

Beyond the cosmetic concerns, recurrent bleedings may lead to functional impairment in patients’ professional and daily life [40]. Prevention of mechanical trauma and destruction of existing telangiectases are the main strategies in treating recurrent epistaxis. Nasal humidification is recommended to limit crust formation and mechanical trauma to the mucosa [41,42]. Alternatively, destruction of telangiectases can be performed, usually by laser therapy [43,44]. Cutaneous telangiectases, the vascular stigmata of HHT [45], are easily identifiable and accessible to treatment. A retrospective survey of patients treated with endonasal Nd:YAG laser treatment revealed that the duration of the interval between treatment sessions increased with the number of interventions [46]. Nevertheless, some of our patients stated that epistaxis worsened after commencing endonasal treatment (unpublished results from the outpatient clinics of the authors), indicating that this approach is not reliably effective for all patients. 

Sunlight is comprised of a spectrum of light (UV, visible, and infrared) with varying wavelengths. The induction of telangiectases is presumably caused by widely reported damaging UV light (within the shorter wavelengths from 100 to 400 nm) [47]. It remains unclear whether light-induced trauma of longer wavelengths, such as within the infrared region, has as much of an influence in stimulating telangiectasia development. Light from the Nd:YAG laser (wavelength = 1064 nm) is used to destroy a vascular lesion and may lead to more scar formation, antagonizing induction of new telangiectases [48]. There is still evidence lacking if laser treatment could prevent the development of new telangiectases. Hence, before elective laser treatment patients should be informed about a possible long-term deterioration by induction of new telangiectases. Additionally, the concomitant therapy with drugs which might compensate for the genetic loss, e.g., tranexamic acid [49], during the period of wound healing could be discussed with the patients. 

In conclusion, we could demonstrate that in patients with HHT mechanical and light-induced trauma promote the formation of telangiectases. Preventing these traumata might also prevent the formation of new telangiectases. 

### Study Limitations

It should be noted that a limitation of the study was the difficulty in grading both sunlight exposure and strain by manual work. The method chosen (giving general examples for each grade level) was quite vague, and several patients did not fully understand how to self-assess themselves. However, a more quantitative categorization (e.g., estimated intensity combined with mean daily hours of sun light exposure or manual work) proved to be impractical or more complicated based on discussions with patients during the study preparation stage. Additionally, exposition to the discussed factors did not occur under controlled conditions. Other unknown factors might have played a role, too, so that uncertainties in the hypothesized chain of causation remain. 

## Figures and Tables

**Figure 1 jcm-09-01507-f001:**
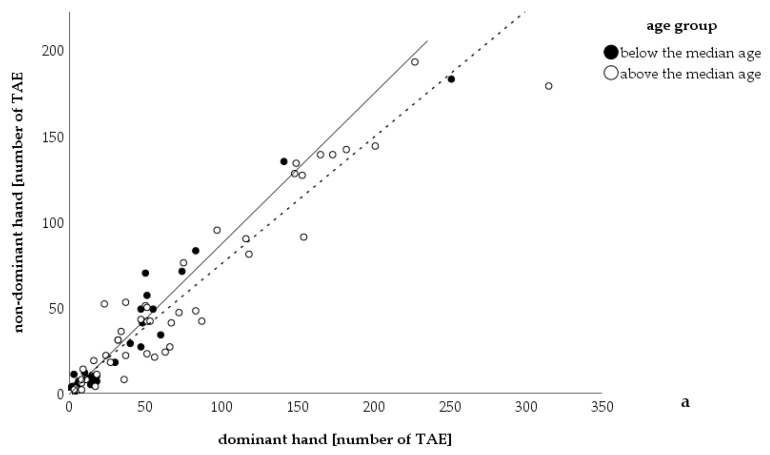
Correlation between number of telangiectases on the dominant and non-dominant hand. Number of telangiectases counted on the dominant hand correlated with the number of telangiectases on the non-dominant hand. The number of telangiectases on the dominant hand was significantly higher than on the non-dominant hand (Wilcoxon rank sum test: *p <* 0.001). The median age was 53 years (minimum: 9 years, maximum 83 years, *n =* 102). The continuous line is the bisectrix, dots lying on this line represent patients’ equal numbers of telangiectases on both hands; the dashed line is the tendency line. TAE: telangiectases.

**Figure 2 jcm-09-01507-f002:**
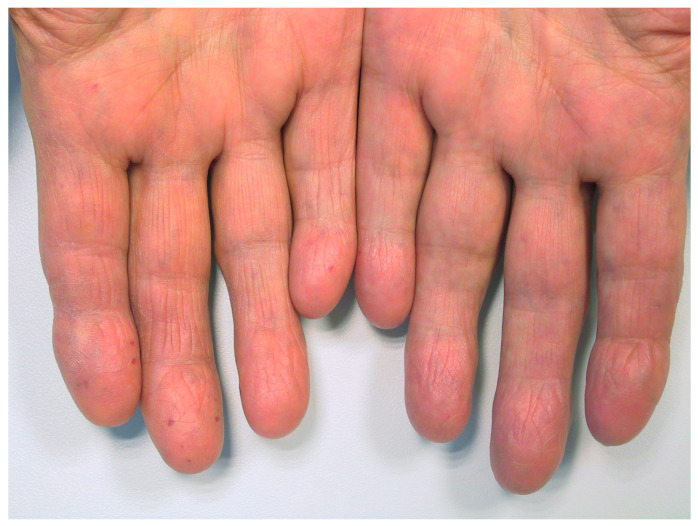
Photo of patient’s hands with more telangiectases on the right dominant hand. The hands of a right-handed female patient of 68 years are shown. Her hands were exposed to relatively low mechanical strain in the past. A total of 36 telangiectases were counted on her right and eight telangiectases on her left hand (only some of them are visible in the photo). She experienced two episodes of bleeding of telangiectases on her right hand, however, never on the left hand.

**Figure 3 jcm-09-01507-f003:**
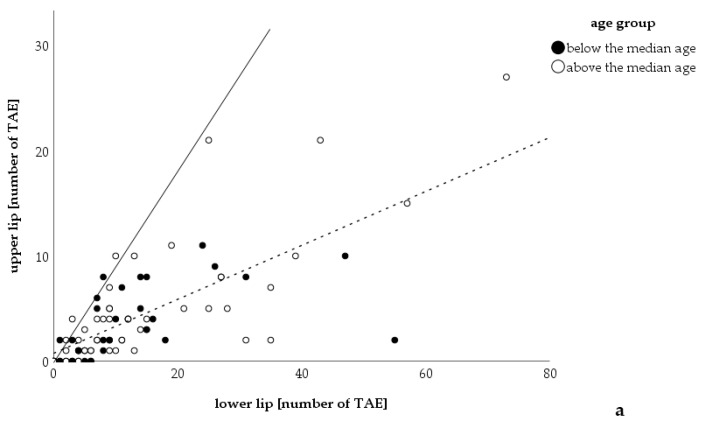
Correlation between number of telangiectases on the upper and lower lip. Number of telangiectases counted on the upper lip correlated with the number of telangiectases on the lower lip (*n* = 86). The number of telangiectases of the lower lip was significantly higher than the number on the upper lip (Wilcoxon rank sum test: *p <* 0.001; mean number of TAE: upper lip *=* 4 TAE, lower lip = 13 TAE). The median age was 54 years (minimum: 9 years, maximum 83 years, *n =* 86). The continuous line is the bisectrix, dots lying on this line represent patients with equal numbers of telangiectases on both lips; the dashed line is the tendency line. TAE: telangiectases.

**Figure 4 jcm-09-01507-f004:**
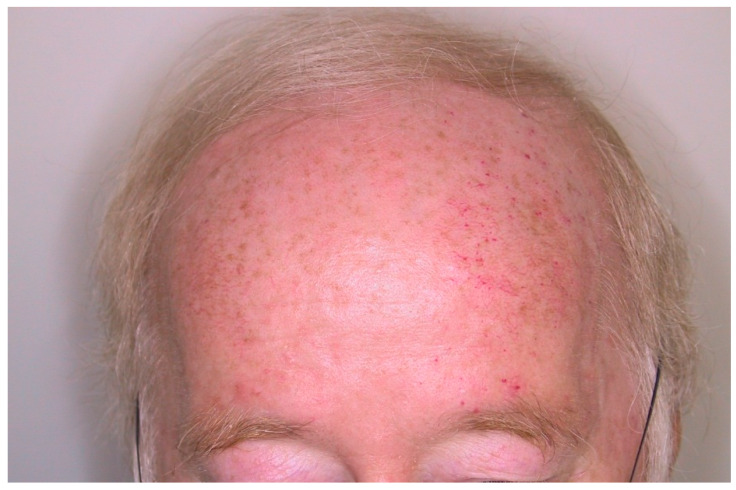
Photo of a patient’s head with more telangiectases on the left forehead after sun exposure. This patient (65 years old) reported that he usually sat on the same bench in the afternoon. The sun used to shine from the left side. He did not report on any mechanical stress in that region.

**Table 1 jcm-09-01507-t001:** Patients classified by handedness and number of telangiectases.

	Right Handed (*n*)	Ambidextrous (*n*)	Left Handed (*n*)	Total
more TAE on dominant hand (*n*)	67	0	2	69
equal number of TAE (*n*)	13	2	1	16
more TAE on non-dominant hand (*n*)	14	2	1	17
Total	94	4	4	

*n*: number of patients (total: 102), TAE: telangiectases, two patients were ambidextrous and had an unequal number of TAE on the hands.

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
