# Peer review of "Trauma Can Induce Telangiectases in Hereditary Hemorrhagic Telangiectasia"

_jcm, 2020, doi:10.3390/jcm9051507_

Round 1
Reviewer 1 Report
This paper contains data that were first presented as abstracts several years ago. The data consists in counting the number of telangiectases on lips and hands and a survey of patient's answers to define environmental factors affecting the generation of telangiectases.
The conclusions are that exposure to trauma in the form of mechanical (manual activity) and sun-induced lead to increased number of telangiectases on hands. A positive correlation is shown for sun light effect on lower vs upper lip but this was hard to quantify although suggestive. The sun exposure (likely UV light) effect on facial telangiectases is based on one patient image and how he sits on his balcony, and no quantitation was in fact performed. So the UV data is somewhat weak.
The recent study of Marchulk's group showing bi-allelic loss of function due to a somatic mutation in addition to the germ-line mutation in individual telangiectases of either HHT 1 or HHT2 patients should have been discussed in this paper. This is the "classical second hit" theory envisaged for cancer by Knudson. A more general two hit model proposed for various diseases argue that factors other than genetic act as second hits, such as trauma which may affect different genes in a given pathway. The authors should discuss how they interpret the two hit model in their case and what their data suggest.
-The following sentence in the abstract is incomplete and needs correction "To assess the influence of 23 environmental trauma (mechanical or light induced) on the number of telangiectases in patients with HHT."
-in figure 3: sentence should be corrected. It is significantly higher or statically significant
"The number of telangiectases of the lower lip is statistically significant higher than the number on the upper lip. "
Reviewer 2 Report
This manuscript focused on the relation between telangiectasias formation and sun exposure and/or mechanical stress. The study is easy to follow and easy to understand, and the statistical analysis seems to be well done. It needs little improvement on the references section in the introduction part and some other comments through the text and Figures.
Abstract, line 21: HHT does not only results in (muco)cutaneous telangiectasias, also in AVMs and internal and bleedings. So from a medical point of view could be good to add this information in the Abstract section.
Abstract, line 24-25: The number of telangiectasias were counted to asses the influence of the environmental trauma. One thing implies the other, so It is better to eliminate the ‘point’ between the two sentences and separate them with a ‘comma’.
Line 34. The results of this study could have a prevention application but not so much as therapeutic approach. It will be better to avoid that sentence “therapeutic approach” in the Abstract.
Line 48: change “rupturing” for “rupture”
Line 54: when talking about TGFb signaling pathway and ALK1 and Eng proteins as the more frequent protein affected, add some references about it. The reference number 10, is only about a mutation in ALK1 gene. There are other paper which are more important, talking about the mechanism of TGFb signaling and Eng and ALK1 proteins together. Please, add other references to improve this part.
Line 55, Reference 10, the same that last comment. I think it is not the best reference if talking about both proteins, try to find other papers, this one is ok for patients and it is ok to have it here, but it has been seeing also in mice models during development which shows clearly than in homozygosity after p10-12 the mice don’t survive.
Line 56: Please, describe the “second hits” or “insults” and the theory to add more information in the introduction part. The same that last comment, there are some other papers which could be important to add.
Line 57: Please relocate the reference number 12 in the paper. This reference talks about modifier genes and how they can influence the AVMs formation in a mouse model and HHT severity, but not about mechanical induction of AVMs.
Line 77, Part of Study approval, if approve document has a number, please cite it.
Line 89, if the patients are left or right-handed is not relevant for this study. Eliminate the sentence.
Figure 1. If there is some differences between ages and number of telangiectasias as it is clearly written in the text, could it be possible to have a graph showing that? Or could it be added in the graph already done, marking differently the samples depending of age-range? Equal number of TAE in both hands is more normal when younger? If so, could be nice to add it in the text. In the graph it could be also nice to see the tendency line, which can give a more accurate idea of the results.
Table 1: Please define TAE.
Line 116-118. It is important to say there is a tendency, as it seems not to be statistically significant
Figure 2, the explanation of the figure is the same as the title, please add if the patient is older or/and if it is correlated with more mechanical effort in that example.
Figure 3. Similar comment than in Figure 2. Although there is no correlation with age in this case, it will be nice if the data could be represented in the same graph but grouped in age-range decided. Please adjust better the bisectrix line, as it seems not to be adjusted at the same number of both telangiectasias in the axes. If possible, add also the tendency-data line.
Line 129-133. Although it is clear the influence of sun exposure on the left forehead, do you have any more patients in the same situation or after big time exposure to sun which could add more robustness to this part of the study? If this is the case, please add it.
Line 145-146. This affirmation is ok in the discussion section, and can be shown better if as recommended above, in Figures-suggestions, the different age ranges are also shown in the figures-graphs.
Line 157. Please avoid the use of the word “strong” as it is not the only factor in a person which could influence the telangiectasias formation.
Line 164-165, change the order. “lower lip is exposed to the sun more” for “lower lip is more expose to the sun”
Line 193, it can be added more examples of topical treatments and some references as timolol and propranolol.
Line 194, final of discussion section. Please provide a small paragraph summarizing the more relevant study findings and why could be important for the scientific-medical-patient community as a preventive measurement.
Reviewer 3 Report
In this manuscript, Geisthoff et al. investigate the effect of environmental traumas on the number of telangiectases present in patients with Hereditary Hemorrhagic Telangiectasia (HHT). They conclude that mechanical- and ultraviolet-induced trauma can increase the formation of telangiectases, suggesting new potential prophylaxis measures for HHT patients.
Overall assesment
This is a short and straightforward manuscript that provides interesting results supporting clinical evidence on the existence of an environmental second hit in HHT. The paper is well-structured and reads well, although some clarifications are needed regarding the previous literature on the pathogenic mechanism in HHT.
Specific points.
- The Introduction/Discussion section should be enriched by further discussing the underlying molecular mechanism involved in HHT pathogenesis. For example, the authors should further explain the hypothesis of haploinsuficiency/monoallelic loss of function combined with a second hit, by citing and commenting some relevant/recent reviews on the subject (PMID: 20870325; PMID: 28796572).
- Page 5, lines 149-152. “In several other types of vascular malformations, such as inherited subsets of venous malformations and cerebral cavernous malformations, complete bi-allelic, localized loss of function of a gene of interest is required for formation of vascular lesions [16-18]. Though this hypothesis may apply to HHT, there has not been confirmation of such an event in patients.”
The authors must update this information as the group of Dr. Marchuk has recently reported the existence of low-frequency somatic mutations in HHT telangiectases, suggesting the existence a of biallelic loss of expression of the HHT genes ENG and ACRL1 (PMID: 31630786). This article should be cited and its implications commented in the context of the KO HHT animal models (references 11,19-23 in the manuscript), as well as the environmental second hits in HHT, as those described in this manuscript.
- The authors may wish to expand the Discussion section by citing and commenting: i) A report by Baeyens et al. on the role of fluid shear stress mechano-transduction as a second hit in HHT (PMID: 27646277), and ii) A recent review by Hiepen et al on “Biomechanical Stress Provides a Second Hit in the Establishment of BMP/TGFβ-related Vascular Disorders” (PMID: 32043077)
Round 2
Reviewer 1 Report
In your response to reviewers, you cited:
"To our opinion it still remains unclear, if a genetic second hit is necessary for the development of all telangiectases in HHT and if environmental insults like trauma just can be a starting factor or if these insults together with a haploinsuficciency might be sufficient in some cases. We included this statement. "
However, this is not quite what you stated in the discussion and you propose a third hit model which is quite an extrapolation from the above statement. It could be either a second mutation or perhaps a trauma as second hit as shown in mouse models and as referred to for several diseases where a more general notion of a second hit is proposed.
A few typos:
Abstract: Hereditary hemorrhagic telangiectasia (HHT) is an autosomal dominant disease of the 20 fibrovascular tissue resulting in visceral vascular malformations
Line 156: a 65-year-old male patient, who exhibited
